# Reclassification of Treatment Strategy with Fractional Flow Reserve in Cancer Patients with Coronary Artery Disease

**DOI:** 10.3390/medicina58070884

**Published:** 2022-07-01

**Authors:** Jin Wan Kim, Tariq J. Dayah, Awad Javaid, Dominique J. Monlezun, Dinu V. Balanescu, Teodora Donisan, Kaveh Karimzad, Abdul Hakeem, David L. Boone, Nicolas Palaskas, Juan Lopez-Mattei, Peter Y. Kim, Jean-Bernard Durand, Juhee Song, Serban M. Balanescu, Eric H. Yang, Joerg Herrmann, Konstantinos Marmagkiolis, Konstantinos Toutouzas, Nils P. Johnson, Cezar A. Iliescu

**Affiliations:** 1Department of Cardiology, The University of Texas Health Science Center at Houston, Houston, TX 77030, USA; tariq.dayah@gmail.com (T.J.D.); david.l.boone@uth.tmc.edu (D.L.B.); nils.johnson@uth.tmc.edu (N.P.J.); ciliescu@mdanderson.org (C.A.I.); 2Department of Cardiology, Kirk Kerkorian School of Medicine, University of Nevada Las Vegas, Las Vegas, NV 89154, USA; awadiqbaljavaid@gmail.com; 3Department of Cardiology, The University of Texas MD Anderson Cancer Center, Houston, TX 77030, USA; dominique.monlezun@gmail.com (D.J.M.); dinu.balanescu@yahoo.com (D.V.B.); teodora.donisan@gmail.com (T.D.); kkarimzad@mdanderson.org (K.K.); nlpalaskas@mdanderson.org (N.P.); juan.lopezmattei@leehealth.org (J.L.-M.); pkim123@gmail.com (P.Y.K.); jdurand@mdanderson.org (J.-B.D.); 4Robert Wood Johnson Hospital, Rutgers University, New Brunswick, NJ 08901, USA; abdul.hakeem@gmail.com; 5Department of Biostatistics, The University of Texas MD Anderson Cancer Center, Houston, TX 77030, USA; jsong1@mdanderson.org; 6Department of Cardiology, Elias Emergency University Hospital, Carol Davila University of Medicine and Pharmacy, 050474 Bucharest, Romania; sbalanescu@gmail.com; 7Department of Medicine, University of California Los Angeles, Los Angeles, CA 90095, USA; eric.yang@gmail.com; 8Department of Cardiovascular Diseases, Mayo Clinic, Rochester, MN 55905, USA; joerg.hermann@gmail.com; 9Pepin Heart Institute Florida Hospital, Tampa, FL 33613, USA; c.marmagiolis@gmail.com; 10First Department of Cardiology, Athens Medical School, Hippokration Hospital, 11527 Athens, Greece; ktoutouz@gmail.com

**Keywords:** cardio-oncology, coronary artery disease, fractional flow reserve, percutaneous coronary intervention, quantitative coronary angiography

## Abstract

*Background and Objectives*: Cancer and coronary artery disease (CAD) often coexist. Compared to quantitative coronary angiography (QCA), fractional flow reserve (FFR) has emerged as a more reliable method of identifying significant coronary stenoses. We aimed to assess the specific management, safety and outcomes of FFR-guided percutaneous coronary intervention (PCI) in cancer patients with stable CAD. *Materials and Methods*: FFR was used to assess cancer patients that underwent coronary angiography for stable CAD between September 2008 and May 2016, and were found to have ≥50% stenosis by QCA. Patients with lesions with an FFR > 0.75 received medical therapy alone, while those with FFR ≤ 0.75 were revascularized. Procedure-related complications, all-cause mortality, nonfatal myocardial infarction, or urgent revascularizations were analyzed. *Results*: Fifty-seven patients with stable CAD underwent FFR on 57 lesions. Out of 31 patients with ≥70% stenosis as measured by QCA, 14 (45.1%) had an FFR ≥ 0.75 and lesions were reclassified as moderate and did not receive PCI nor DAPT. Out of 26 patients with <70% stenosis as measured by QCA, 6 (23%) had an FFR < 0.75 and were reclassified as severe and were treated with PCI and associated DAPT. No periprocedural complications, urgent revascularization, acute coronary syndromes, or cardiovascular deaths were noted. There was a 22.8% mortality at 1 year, all cancer related. Patients who received a stent by FFR assessment showed a significant association with decreased risk of all-cause death (HR: 0.37, 95% CI 0.15–0.90, *p* = 0.03). *Conclusions*: Further studies are needed to define the optimal therapeutic approach for cancer patients with CAD. Using an FFR cut-off point of 0.75 to guide PCI translates into fewer interventions and can facilitate cancer care. There was an overall reduction in mortality in patients that received a stent, suggesting increased resilience to cancer therapy and progression.

## 1. Introduction

A growing population of cancer patients is diagnosed with coronary artery disease (CAD) due to the combination of improved cancer treatments, prolonged longevity, and common risk factors. For interventional cardiologists caring for cardio-oncology patients [1], several unique considerations apply regarding percutaneous coronary intervention (PCI). These patients frequently require non-cardiac interventions (e.g., tumor resection and other invasive diagnostic procedures), thus influencing dual antiplatelet therapy (DAPT) duration [2]. 

Although the accurate diagnosis of CAD that would benefit from PCI is always important, it is especially relevant in cancer patients, in whom a commitment to DAPT even for a short period of time can complicate cancer care. A vast body of evidence based clinical research supports coronary lesion assessment using fractional flow reserve (FFR) over angiography in the general population [3,4]. An FFR-guided approach specifically for cancer patients seems logical, but is not currently reported in the literature. Our study aims to bridge that gap by describing reclassification through FFR-guided strategy and outcomes in a population of cancer patients with stable CAD. Since the threshold to revascularize should be high, we have increased specificity and considered the FFR of 0.75 from the initial DEFER trial [4] as a reasonable cut-off point for revascularization decision in cancer patients.

## 2. Materials and Methods

### 2.1. Data Sources and Study Population

All cancer patients with symptomatic stable coronary artery disease who underwent FFR assessment between August 2008 and September 2016 were included. Evaluation of stable angina was performed via noninvasive imaging, though due to the typical work-flow of a tertiary center, there was heterogeneity of data (Exercise/dobutamine stress echo, adenosine/lexiscan nuclear stress test, cardiac PET) and initial work-up performed in outside various institutions in addition to low numbers. Patients were evaluated to have a life expectancy of at least 6 months and preferably 1 year survivorship by the oncology team. The study protocol conformed to the ethical guidelines of the 1975 Declaration of Helsinki as reflected in the approval by the MD Anderson institutional review board and those of the American Physiological Society (protocol code DR10-0812, 18 November 2011) Subject-level consent was waived, given the study’s retrospective design and the fact that FFR was provided according to standard clinical care.

Baseline demographic and clinical information was extracted from a chart review. The severity of each coronary lesion was assessed by both quantitative coronary angiography (QCA) and FFR. In all procedures, intravenous adenosine was infused at 140 µg/kg/min for hyperemia [5]. 

### 2.2. FFR-Guided Treatment and Follow-Up

The indication for coronary angiography was stable angina. Coronary lesions were classified into 2 groups based on severity as evaluated by QCA: a group with severe (≥70%) stenosis and a group with intermediate (50–70%) lesions. If the initial severity was changed after FFR assessment, lesions were considered reclassified. FFR protocol was performed as supported by long-term clinical outcomes from a seminal FFR randomized trial in a general population [4], as well as by the agreement between FFR and non-invasive testing [6]. In this patient population, lesions with FFR ≥ 0.75 were treated with optimal medical management (OMT), while lesions showing an FFR < 0.75 were treated with OMT and revascularization, except for the rare cases with an immediate medical contraindication. When multiple coronary lesions were present, the most severe/clinically relevant lesion was used for analysis. PCI was individualized and was performed with both bare-metal stents and drug-eluting stents in an attempt to accommodate the cancer treatment schedule. After PCI, patients received a DAPT regimen consisting of aspirin and a P2Y12 inhibitor (clopidogrel) according to current ACC guidelines, however, given the comorbidities of the population, not all patients were able to tolerate this regimen for the entire period. 

Clinical endpoints: procedure-related complications (bleeding, coronary dissection or perforation, and renal insufficiency) and major adverse cardiovascular events (MACE), defined as myocardial infarction, heart failure, percutaneous cardiac intervention, coronary artery bypass grafting, cardiac death, and overall mortality were obtained from chart review. Myocardial infarction (MI) was defined and assigned according to current American College of Cardiology/American Heart Association (ACC/AHA) guidelines. 

### 2.3. Statistical Analysis

Analysis was performed using SAS version 9.4 (SAS Institute, Cary, NC, USA). Patient characteristics were summarized with descriptive statistics for the entire group, by overall survival status, by 12 months survival status, by revascularization, and by MI. Continuous variables were compared between groups utilizing the Wilcoxon rank-sum test. Categorical variables were compared utilizing Chi-square test or Fisher’s exact test. Univariate Cox proportional hazards regression was used to identify factors that are significantly associated with the risk of death. A *p*-value of less than 0.05 indicated a statistical significance. 

## 3. Results

Our population included 57 patients with demographic and clinical characteristics detailed in Table 1. The procedure was technically successful in all cases, without any periprocedural complications (bleeding, coronary dissection or perforation, and renal insufficiency). Most FFR assessments were performed on the left anterior descending artery lesions (68.4%). We identified 23 (40.35%) lesions with FFR < 0.75, all of which were treated with PCI, and 34 (59.64%) lesions with FFR ≥ 0.75, of which 12 were in the “grey zone” of 0.75–0.8 and were treated with OMT. The stents used were 10 (43.5%) BMS and 13 (56.5%) DES to accommodate cancer treatment. Out of 31 patients with ≥70% stenosis as measured by QCA, 14 (45.1%) had an FFR ≥ 0.75 and lesions were reclassified as moderate and did not receive PCI or DAPT (Figure 1). By contrast, out of 26 patients with <70% stenosis as measured by QCA, 6 (23%) had an FFR < 0.75 and were reclassified as severe and were treated with PCI and the associated DAPT.

Over a median of 43 months (95% confidence interval (CI) 31–53 months) of follow-up, a total of 22 (38.6%) deaths occurred. From the entire group of patients, there was one (1.8%) death due to a cardiovascular cause. No urgent revascularization or acute coronary syndromes were observed.

Patient characteristics are summarized by survival status along with univariate Cox regression analysis results in Table 2. At 12 months, the only borderline significant factors associated with a slightly better survival were male gender (hazard ratio (HR): 0.38, 95% CI, 0.13–1.13, *p* = 0.08) and receiving a stent (HR: 0.34, 95% CI, 0.10–1.09, *p* = 0.07). When observed for overall survival, patients who received a stent (a decision guided by FFR assessment) showed a significant association with decreased risk of all-cause death (HR: 0.37, 95% CI, 0.15–0.90, *p* = 0.03) (Figure 2A). We further analyzed two subgroups of the patients who did not receive a stent: those with an FFR between 0.75 and 0.8 (11 patients, 19.29%) and those with an FFR > 0.8 (23 patients, 40.35%). Overall survival was not significantly different between these two groups (HR: 0.18, 95% CI, 0.42–3.45, *p* = 0.73).

There was no delay in cancer care (chemotherapy, radiation, or surgery) in medically or PCI-managed patients. No immediate revascularization, acute coronary syndromes, or cardiovascular deaths were noted, but there was a 22.8% mortality at one year, all cancer related (disease progression or cancer therapy related complications). 

Our study found a trend of improved overall survival in patients who had hemodynamically significant lesions based on FFR (≤0.75) (Figure 2B). However, the differences were not statistically significant (Log-rank *p*-value = 0.44) regarding mortality of patients in the four groups based on the combination of FFR and QCA assessment of stenosis severity. 

## 4. Discussion

To the best of our knowledge, this study provides the first dedicated report of the outcomes of FFR-guided treatment in cancer patients. While not randomized, several observations from our study reflect themes from the broader FFR literature. For example, we observed a substantial bidirectional reclassification between the two groups (intermediate to severe and vice versa) when lesion severity was further investigated with FFR. Therefore, angiographic-based PCI in stable patients without FFR confirmation leads to unnecessary revascularizations [7] that can be especially harmful for cancer patients requiring invasive non-cardiac interventions (e.g., tumor resection) or chemotherapy. Despite the increased thrombotic risk in cancer patients [8], none of the lesions deferred or stented presented thrombotic complications. Our high technical success rate and lack of adverse events confirms other observations that FFR is a safe and reliable technique [9]. Finally, the large number of non-cardiac deaths was more apparent in our oncology population than in the general CAD population, emphasizing competing risks from comorbid conditions [10]. 

In our studied groups, we could not find any statistically significant differences in mortality based on the combination of FFR and QCA. This finding, as well as the trend of improved survival in patients who had hemodynamically significant lesions based on FFR alone, might suggest that FFR is an appropriate tool to decide the therapeutic approach in this group of patients. 

The cancer type also seems to impact survival, as observed outcomes were worse in patients with hematologic (versus solid) malignancies. Cancer patients have heterogeneous clinical presentations and comorbidities, with unpredictable complications and the majority with hemodynamic impact; we believe this is the reason why stented patients showed better survival. It is possible that the trend of improved survival following revascularization in patients with severe coronary disease is due to an increased resilience to the challenges of cancer therapy.

Given the continuous relationship between the FFR value and subsequent outcomes seen in many studies [3,11,12] and the heterogeneity of clinical presentations, the decision to revascularize CAD must incorporate all available information. This truism applies especially to the cancer patient, requiring that the interventional cardiologist have knowledge of oncologic disease status as well as ongoing or planned therapies such as surgery, chemotherapy, or radiation. The current guidelines recommend DAPT after PCI with stent implantation for at least 6–12 months, with an emerging trend toward individualization of treatment duration [13]. This adds to the complexity of management in this already high-risk population.

This type of clinical synthesis, coupled with the innately higher risk and the comorbid conditions of cancer patients, has led us to adopt a general FFR threshold of ≤0.75, as supported by long-term follow-up from a randomized trial [4], as well as the agreement between FFR and non-invasive testing [6]. Although there is evidence supporting an even lower FFR threshold of ≤0.65 for PCI [14], cancer therapy is a “roller-coaster”, where anemia, tachycardia, hypotension, hypertension, dehydration and pain are not uncommon drivers of supply/demand mismatch and type II MI. Part of cardiovascular optimization is improving patients’ resilience to these clinical challenges; therefore, we aimed to avoid using the ≤0.75 value. The larger mortality benefit seen in patients after receiving a stent may also reflect ways in which intervention enhances patients’ resilience against these type II MI events.

We did not identify a significant increase in mortality in patients with FFR between 0.75 and 0.8 compared to >0.8 who did not undergo PCI, suggesting that PCI deferral is safe even using this threshold. Prior work has suggested a threshold of 0.79 for major adverse cardiac events that includes revascularizations, but a lower threshold of 0.64 for hard endpoints alone [12]. Given the large number of cancer-related mortalities in our population, adopting a lower FFR threshold for revascularization may be appropriate but does not extend to other populations. This result also raises the question of the benefit of performing PCI versus OMT in patients with a life expectancy of <1 year. Ongoing studies such as GzFFR (clinicaltrials.gov NCT02425969) will shed more light on this topic.

Our patients derived a benefit from BMS or DES implantation. Another method that is not available in the United States of America is PCI with a drug-eluting balloon. Drug-eluting balloons may provide advantages over stent implantation by avoiding the risk of stent thrombosis and decreasing the dependence on DAPT [15]. This could represent an alternative treatment in cancer patients with contraindications to DAPT, although data are scarce in this regard.

This study represents a first step towards the determining optimal physiologic assessment of coronary lesions in cancer patients with stable CAD. Further research should focus on a more extensive use of non-invasive physiologic assessment with computed tomography FFR (CT-FFR). This approach could be of particular benefit in cancer patients with thrombocytopenia, who might otherwise be deferred from cardiac catheterization due to a theoretical concern for bleeding [16]. However, CT-FFR has not yet become the standard of care and no data currently exist on the use of this modality in cancer patients. 

## 5. Conclusions

Our FFR approach was able to minimize the effects of cardiovascular mortality and morbidity during the course of patients’ cancer treatments. FFR-guided treatment should be considered in cancer patients with stable coronary artery disease and intermediate or severe lesions. As there are differences in lesion severity when evaluated by QCA or FFR, a reclassification of lesion severity based on FFR to avoid unnecessary stenting should be considered. Using an FFR cut-off point of 0.75 to guide PCI translates into fewer interventions and can facilitate cancer care. There was an overall reduction in mortality in patients that received a stent, suggesting increased resilience to cancer therapy and progression. While the number of complications is reduced even in this high-risk population, advanced contemporary imaging modalities such as CT-FFR can be considered when the validation process is complete. As stent platforms have improved and complications rates have decreased, we also recognize a trend towards a complete revascularization approach that will be the focus of further studies.

## Figures and Tables

**Figure 1 medicina-58-00884-f001:**
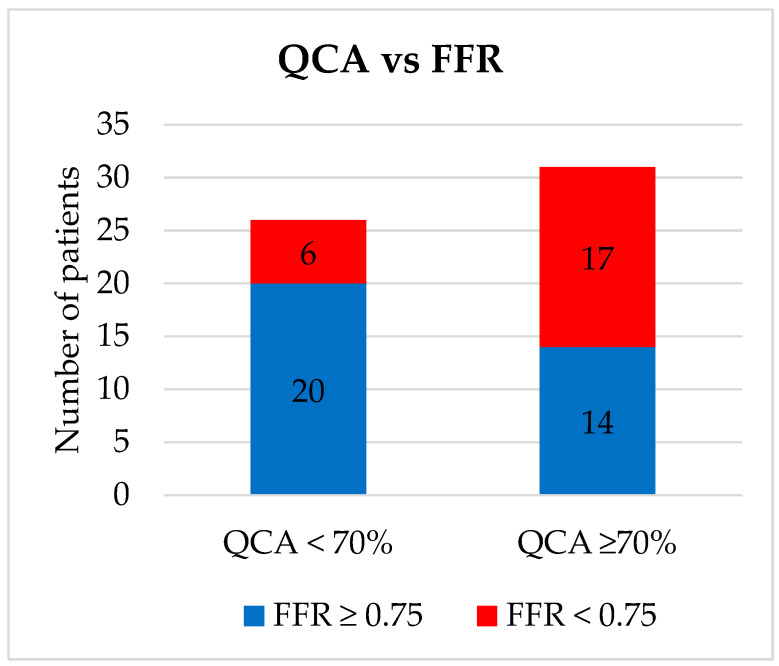
Coronary stenosis severity assessment via quantitative coronary angiography versus fractional flow reserve. QCA = Quantitative coronary intervention; FFR = Fractional Flow Reserve.

**Figure 2 medicina-58-00884-f002:**
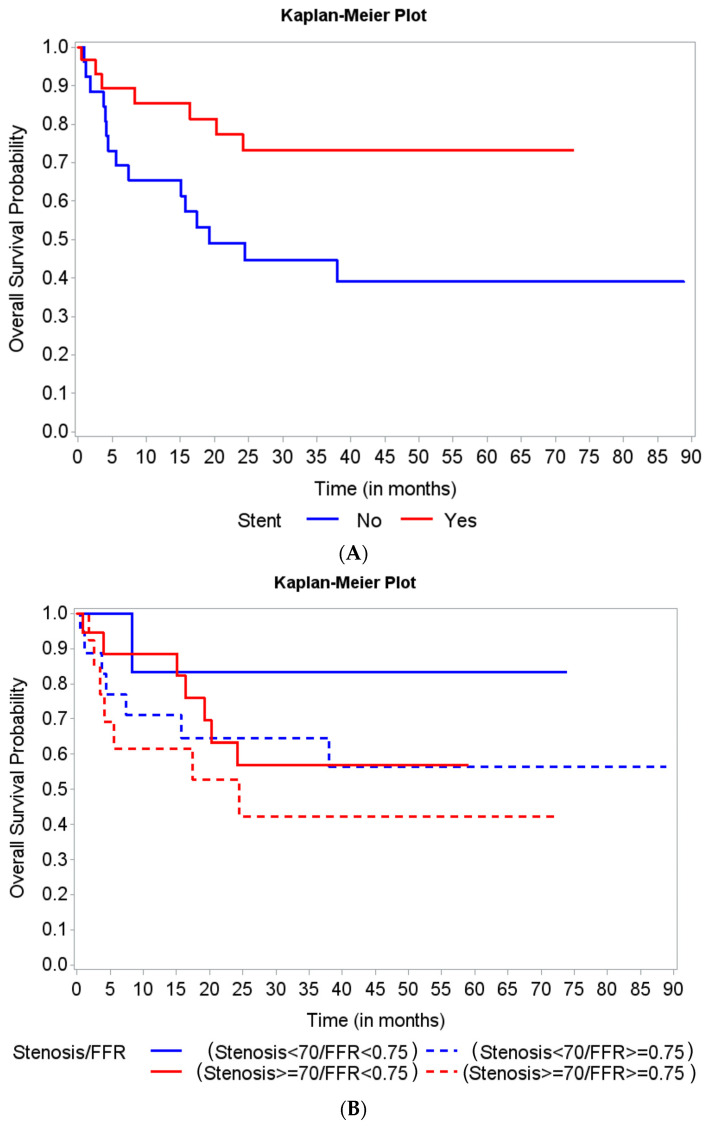
Kaplan Meier curves for overall survival based on stent implantation and lesion severity assessed by fractional flow reserve (FFR) and quantitative coronary angiography (QCA). (**A**) Patients who received a stent (a decision guided by FFR assessment) showed a significant association with a decreased risk of all-cause death. (**B**) The overall survival did not show any significant differences among the 4 groups of patients, based on the lesion severity assessment type.

**Table 1 medicina-58-00884-t001:** Descriptive statistics for the entire group (n = 57).

Variables	Mean ± SD or Count (%)
Age (years)	64.8 ± 9.92
Female	16 (28.1%)
Male	41 (71.9%)
Cardiovascular risk factors	
Hypertension	41 (71.9%)
Hyperlipidemia	34 (59.6%)
Smoking	25 (43.9%)
Family history of coronary artery disease	12 (21.1%)
Malignancy type	
Solid	43 (75.4%)
Hematologic	14 (24.6%)
Hemoglobin (g/dL)	11.8 ± 2
Platelet number (K/µL)	193.5 ± 78.28
Creatinine (mg/dL)	1.13 ± 0.94
Coronary lesions	
Left anterior descending artery	39 (68.4%)
Left circumflex artery	3 (5.3%)
Left main coronary artery	4 (7%)
Right coronary artery	10 (17.5%)
Ramus	1 (1.8%)
Stenosis severity (%)	65.88 ± 9.92
<70%	25 (43.9%)
≥70%	32 (56.1%)
Fractional flow reserve	0.77 ± 0.12
Percutaneous coronary intervention with stenting	
No	34 (59.6%)
Yes	23 (40.4%)
Bare-metal stent	10 (43.5%)
Drug-eluting stent	13 (56.5%)
Death within 12 months	13 (22.8%)
Cardiovascular mortality	1 (1.8%)

**Table 2 medicina-58-00884-t002:** Patient characteristics by survival status, univariate Cox regression (censored at last follow-up).

		Survival at 12 Months				Overall Survival			
		Alive (N = 44)	Dead (N = 13)	HR (95% CI)	*p*	Alive (N = 35)	Dead (N = 22)	HR (95% CI)	*p*
Age (years)		64.52 ± 10.71	65.76 ± 6.84	1.00		62.84 ± 10.83	67.92 ± 7.47	1.03 (0.99–1.08)	0.11
Gender	Female	10 (22.7%)	6 (46.2%)	1.00		9 (25.7%)	7 (31.8%)	1.00	
	Male	34 (77.3%)	7 (53.8%)	0.38 (0.13–1.13)	0.08	26 (74.3%)	15 (68.2%)	0.77 (0.31–1.89)	0.57
Hypertension	Yes	32 (72.7%)	9 (69.2%)	0.83 (0.26–2.70)	0.76	24 (68.6%)	17 (77.3%)	1.19 (0.44–3.23)	0.73
Hyperlipidemia	Yes	27 (61.4%)	7 (53.8%)	0.76 (0.25–2.25)	0.62	22 (62.9%)	12 (54.5%)	0.78 (0.34-1.80)	0.56
Smoking	Yes	20 (45.5%)	5 (38.5%)	0.76 (0.25–2.32)	0.63	15 (42.9%)	10 (45.5%)	1.16 (0.50–2.69)	0.73
Family history of coronary artery disease	Yes	10 (22.7%)	2 (15.4%)	0.69 (1.53–3.12)	0.63	9 (25.7%)	3 (13.6%)	0.63 (1.88–2.14)	0.46
Stent	Yes	27 (61.4%)	4 (30.8%)	0.34 (0.10–1.09)	0.07	24 (68.6%)	7 (31.8%)	0.37 (0.15–0.90)	0.03
Stenosis severity group	<70%	19 (43.2%)	6 (46.2%)	1.00		17 (48.6%)	8 (36.4%)	1.00	
	≥70%	25 (56.8%)	7 (53.8%)	0.93 (0.31–2.75)	0.89	18 (51.4%)	14 (63.6%)	1.50 (0.62–3.59)	0.37
Fractional flow reserve		0.77 ± 0.12	0.77 ± 0.12	1.00		0.78 ± 0.12	0.76 ± 0.13	0.32 (0.01–10.00)	0.51

## Data Availability

The data presented in this study are available on request from the corresponding author. The data is not publicly available for the protection of personal health data.

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
