# Peer review of "Reclassification of Treatment Strategy with Fractional Flow Reserve in Cancer Patients with Coronary Artery Disease"

_medicina, 2022, doi:10.3390/medicina58070884_

Round 1

Reviewer 1 Report

The authors describe results from an FFR-guided treatment strategy in cancer patients with stable coronary artery disease and intermediate or severe lesions. The topic is interesting, the manuscript is well written and objectives, results, and discussion are clearly outlined. Apart from a previously published meeting abstract that presented the main results, this is the first original report of the outcomes of FFR-guided treatment in cancer patients. The obvious limitations relate to the non-randomized design and the small sample size. Thus, the results have to be interpreted with caution. Nevertheless, this reviewer has no specific recommendations for improvements.

Reviewer 2 Report

This article reports a study focusing on 57 patients with cancer who developed stable CAD, and those patients took the treatment strategy with lower cut-off value for FFR. The result showed no periprocedural complications, urgent revascularization, acute coronary syndromes, or cardiovascular deaths and a 22.8% mortality at 1 year, all cancer related. Patients who received a stent by FFR assessment was reported a significant association with decreased risk of all-cause death.

Specific comments

1. In DEFER trial, cut-off value as 0.75 was used for FFR in the study. If this article tried to take more conservative strategy for these cancer patients, why the authors still used 0.75 as FFR cut-off point, not lower value?

2. If there any evaluation of life expectancy for those participants before study?

3. The indication for coronary angiography was stable angina. Was there any noninvasive study (ex.: myocardial perfusion imaging) before CAG/PCI?

4. Would you explain why patients without PCI got lower survival even they might have less severe CAD?

5. About 20% patients died at 12 months. Would that mean further conservative strategy was even more suitable for these cancer patients?

6. The authors worried about that cancer treatment would be affected by too invasive strategy with PCI or DAPT after PCI. But in this study all patients died of cancer even after taking new strategy for CAD intervention. How to explain this point? I didn’t realize the benefit for those cancer patient if taking this new FFR strategy.

7. For patients who receiving PCI, would some patients receive staging PCI if necessary? (I think there would be some bias if you only use the most severe/clinically relevant lesion for analysis, and the clinical relation sometimes needs pre-procedural survey such as noninvasive stress test to differentiate which is the most clinically relevant lesion; is there any data you could offer?)

Author Response

Reviewer 2

1. In DEFER trial, cut-off value as 0.75 was used for FFR in the study. If this article tried to take more conservative strategy for these cancer patients, why the authors still used 0.75 as FFR cut-off point, not lower value?
Response: We appreciate the reviewers’ input. Anemia, tachycardia, hypotension, hypertension are part of the “cancer treatment” rollercoaster where disease or treatment related dehydration, sepsis, pain can drive mismatch supply/demand and type II MI. Part of CV optimization is improving patients resilience to these obstacles, therefore we opted for a midline strategy in this initial study. How an even lower FFR cut-off value might affect patient outcomes would be an interesting follow up study.
We have added this detail in paragraph 206-221.

2. Is there any evaluation of life expectancy for those participants before study?
Response: All patients have to have at least 6 months and preferably 1 year survivorship documented by the oncology teams. 
We have added this methods detail in paragraph 77-83

3. The indication for coronary angiography was stable angina. Was there any noninvasive study (ex.: myocardial perfusion imaging) before CAG/PCI?
Response: Patients had noninvasive imaging, though due to typical work-flow of a tertiary center we had heterogeneity of data (exercise/dobutamine stress echo, adenosine/lexiscan nuclear stress test, cardiac PET) and initial work-up performed  in outside various institutions in addition to the low numbers, so we decide to keep the focus of the manuscript on the invasive hemodynamic coronary assessment. 
We have added this methods detail in paragraph 77-83.

4. Would you explain why patients without PCI got lower survival even they might have less severe CAD?
Response: We thank the reviewer for the excellent comments. As we mentioned in the reply to question 1, the lower survival of patients without PCI suggests that potentially a cut-off of 0.8 might be considered in this patient population with likely anemia, tachycardia, and hypotension to improve their resilience to disease, treatment and complications. As such, higher survival rates after stent placement may reflect ways in which intervention enhances patient resilience against potential type II MI events.
We have added a clarifying statement in paragraph 240-242. Preceded by the explanation in lines 234-239

5. About 20% patients died at 12 months. Would that mean further conservative strategy was even more suitable for these cancer patients?
Response: Oncological patients have additional metrics on top of survival, including quality of life, number on cancer treatment cycles received and unpredictable complications that can shorten the survival. This concern will continue to be a challenge for the research in the field.

6. The authors worried about that cancer treatment would be affected by too invasive strategy with PCI or DAPT after PCI. But in this study all patients died of cancer even after taking new strategy for CAD intervention. How to explain this point? I didn’t realize the benefit for those cancer patient if taking this new FFR strategy.
Response: The main goal of interventional cardio-oncology is to annul the effect of cardiovascular mortality and mortality from their cancer treatment. Our FFR approach was able to prevent MACE and prevent cardiovascular death in the 1st year. As cancer care is personalized, all the variables derived from their cancer care will be reflected in all of our CV studies.
We have added a clarifying statement in conclusion paragraph 268-281

7. For patients who receiving PCI, would some patients receive staging PCI if necessary? (I think there would be some bias if you only use the most severe/clinically relevant lesion for analysis, and the clinical relation sometimes needs pre-procedural survey such as noninvasive stress test to differentiate which is the most clinically relevant lesion; is there any data you could offer?)
 Response: We appreciate reviewers excellent perspective. As the stent platforms have improved and complications rate have decreased we have recognized as well in our practice a trend towards a complete revascularization approach that will be the focus of further studies. This paragraph will be added in limitations as well.
We have added a clarifying statement in conclusion paragraph 268-281

Round 2

Reviewer 2 Report

This article aimed to study on FFR-guided strategy of PCI in cancer patients with stable angina, and the results showed positive. Although authors may presume that FFR-guided PCI could be better by angiography-guided PCI in cancer patients, no solid evidence was reported. According to this article, the results suggest a FFR-guided strategy.

Minor comments

In lines 203 – 205, I think authors would like to state that no thrombotic complications after receiving PCI or procedures were deferred. Is that correct? (consider to delete the word“not”?)

Author Response

Reviewer: In lines 203 – 205, I think authors would like to state that no thrombotic complications after receiving PCI or procedures were deferred. Is that correct? (consider to delete the word“not”?)

Response: Thank you for pointing out this error in phrasing. We have corrected the sentence to state, "none of the lesions deferred or stented presented with thrombotic complications"